# Evaluation of ECOSTRESS Thermal Data over South Florida Estuaries

**DOI:** 10.3390/s21134341

**Published:** 2021-06-25

**Authors:** Jing Shi, Chuanmin Hu

**Affiliations:** College of Marine Science, University of South Florida, St. Petersburg, FL 33701, USA; jingshi@usf.edu

**Keywords:** ECOSTRESS, MODIS, sea surface temperature, estuaries, Chesapeake Bay, Lake Okeechobee, Caloosahatchee River Estuary, Florida Bay

## Abstract

Operational coarse-resolution satellite thermal sensors designed for global oceans are often insufficient for evaluating surface temperature of small water bodies. Here, the quality of the thermal data, collected by the ECOsystem Spaceborne Thermal Radiometer Experiment on Space Station (ECOSTRESS), over several South Florida estuaries, Chesapeake Bay, and Lake Okeechobee is evaluated using both in situ and Moderate Resolution Imaging Spectroradiometer (MODIS) Sea Surface Temperature (SST) data. Overall, for SST between ~6 and ~32 °C, ECOSTRESS LST (Land Surface Temperature, used as a surrogate for SST in this study) appears to be slightly underestimated, with the underestimation being more severe at night (−1.13 °C) than during the day (−0.64 °C), in spring and summer (−1.25 ± 1.39 °C) than in autumn and winter (−0.57 ± 0.98 °C), and after May 2019 when two of the five bands failed. The root-mean-square uncertainties of ECOSTRESS SST are generally within 1–2 °C. Spatial analysis further suggests that ECOSTRESS SST covers waters closer to shore and reveals more spatial features than MODIS, with comparable image noise. From these observations, after proper georeferencing and empirical correction of the negative bias, ECOSTRESS SST may be used to evaluate the thermal environments of small water bodies, thus filling gaps in the coarse-resolution satellite data.

## 1. Introduction

Over the past decade, many estuaries in South Florida have been under ecological stress with frequent and recurrent algal blooms and seagrass mortalities, such as cyanobacterial blooms and brown tide blooms in Caloosahatchee River Estuary (CRE) [1,2] and in Florida Bay (FB) [3]. In addition, thermal stress has also been reported, such as the 2010 cold event [4]. Corresponding to these abnormal events, fish kills, seagrass die-offs, coral mortality, and lobster die-offs have been reported [4,5,6,7]. While numerous papers have documented the biological stresses including nutrient enrichment [8,9], fewer studies focused on whether these ecologically important estuaries have been under thermal stress. In order to have such an assessment, a long-term, consistent, temperature record for each estuary needs to be established first, through measurements of sea surface temperature (SST). In this context, following the convention, SST refers to water surface temperature regardless of whether the water is seawater or freshwater.

SST can be measured from both field platforms (such as moorings, drifting buoys, Argo floats, and ships) and satellites (either polar-orbiting or geostationary), each having its own strengths and weaknesses. Field-collected SST data (i.e., in situ SST), after quality control, are accurate, but they are usually sparse in either time or space. Satellite-based SST data have more spatial and temporal coverage for regional or global studies, but their uncertainties need to be quantified and understood. Field-measured SST is often from the top layer of the surface water (a few meters) while satellite-sensed thermal signal comes from the water “skin” (i.e., the top millimeters), although the latter is “calibrated” to match the former using multi-channel linear or non-linear regressions [10,11,12,13]. Furthermore, although operational satellite sensors such as Moderate Resolution Imaging Spectroradiometer (MODIS) or Visible Infrared Imaging Radiometer Suite (VIIRS) have daily coverage of the earth surface, their coarse spatial resolutions (1.1 km nadir resolution for MODIS, 0.75 km nadir resolution for VIIRS) often make it difficult to use for small water bodies such as estuaries. On the other hand, some satellite sensors such as the Advanced Spaceborne Thermal Emission and Reflection Radiometer (ASTER, 90 m) on the Terra satellite or the Thermal Infrared Sensor (TIRS, 100 m) on the Landsat-8 satellite provide finder-resolution thermal data [14,15,16,17], but their site revisit frequency (once every 16 days, and lower after discounting clouds) may be too low to capture short-term SST changes in estuaries.

The launch of the ECOsystem Spaceborne Thermal Radiometer Experiment on the Space Station (ECOSTRESS) in July 2018 may provide a practical solution to the above dilemma, as ECOSTRESS collects thermal data over both land and water with a spatial resolution of 38 m × 68 m and a revisit period of 4–5 days. However, because of its experimental nature, the uncertainties in the ECOSTRESS data products need to be understood before they are used to assess the thermal environments of estuaries.

Hook et al. [18] performed an inflight validation on the thermal channels of ECOSTRESS using several validation sites in Lake Tahoe (CA/NV, USA) and Salton Sea (CA, USA). The correlation between in situ data and at-sensor radiance for the five thermal channels is 0.99, with both preflight absolute radiometric accuracy and in-flight noise equivalent delta temperatures (NEΔT) meeting the mission requirements. Silvestri et al. [19] compared the surface temperatures obtained by ECOSTRESS, ASTER, and TIRS over several Italian volcanic and geothermal areas. The cross-sensor correlations and biases are 0.66 to 0.89 and −2.47 to 0.63 K, respectively. However, despite these two studies, the quality of the ECOSTRESS surface temperature data over estuaries is largely unknown, as the study by Hook et al. [18] is focused on the at-sensor radiance rather than the estimated surface temperature. Furthermore, because small estuaries require accurate geometric positioning of image pixels, it is also essential to evaluate the geometric accuracy of ECOSTRESS surface temperature images. Such an evaluation was mostly ignored in previous studies.

The objective of this study is to evaluate ECOSTRESS surface temperature data collected over South Florida estuaries using in situ measurements and well-calibrated and validated MODIS measurements whenever resolution is not an issue. Because the estuaries are all located in a subtropical setting with a narrow SST range, the study also includes Chesapeake Bay (CB) in order to evaluate ECOSTRESS surface temperature data for a lower temperature range.

## 2. Data and Methods

There is a wealth of literature on the evaluation of remotely sensed SST using either field-measured SST [12,20,21,22,23,24] or SST from another well-calibrated remote sensor [25,26]. In this study, we follow these published works to evaluate ECOSTRESS data using both methods.

### 2.1. In Situ Data and Satellite Data

#### 2.1.1. In Situ Data

In situ surface temperature data for the CB, Lake Okeechobee (LO), CRE, and FB were obtained from the National Data Buoy Center (NOAA NDBC), South Florida Water Management District (SFWMD), and Sanibel-Captiva Conservation Foundation River, Estuary, and Coastal Observing Network (SCCF RECON) (Figure 1 and Table 1). The thermal sensors are mounted on buoys, Coastal-Marine Automated Network (C-MAN) stations, platform-based stations, or other water quality stations. NDBC thermistors are within the first two meters of the water column, while thermistors in other sites are from 0.5 to ~1 m below the ocean surface. The exact depths these thermistors have not been reported as they are also influenced by tides, but measured temperatures are believed to represent the bulk temperatures of the surface water layers rather than the skin temperatures (top millimeters). All data were collected at intervals of less than one hour and had undergone extensive quality control by the data collection agencies/groups.

#### 2.1.2. ECOSTRESS

ECOSTRESS has a TIR multispectral radiometer installed on the Japanese Experiment Module-Exposed Facility (JEM-EF) on the International Space Station (ISS), which measures the radiance in five spectral channels between ~8 and ~12 µm (centered at 8.29, 8.78, 9.20, 10.49, and 12.09 µm), with an additional band at 1.60 µm for geolocation and cloud detection. ECOSTRESS collects data within a swath width of 402 km and with NEΔT < 0.1 K (reference at 300 K) in all five TIR bands [27,28]. After proper atmospheric correction, the surface spectral radiance was used to retrieve the Land Surface Temperature and Emissivity (LST&E) based on the Temperature Emission Separation (TES) algorithm [29,30,31]. Here, ECOSTRESS LST is used as a surrogate of SST, and the two terms are used interchangeably in this text because our focus is on estuaries and inland waters. Due to malfunctioning of the mass storage unit (MSU) on 14 March 2019, only three bands (8.78, 10.49, and 12.09 µm) were available after 15 May 2019, with the other two bands being filled with dummy values. The total RMS errors in the retrieved LST data from the 3-band data were ~1.5 K, as compared to ~1 K from the 5-band data [31].

All ECOSTRESS data are available from the NASA Land Process Distribution Active Archive Center (LPDAAC). The Level-2 product, ECO2LSTE, provides the LST, emissivity (E), and corresponding quality control (QC) with a spatial resolution of approximately 70 × 70 m. QC flags, recorded in digital bits, were used in this study to exclude low-quality data. The value of bits 1&0 = 00 in the QC flags indicates the best quality from the clearest sky as determined from a series of cloud tests, which include morphological tests that are used to flag and discard pixels in the “holes” between clouds in order to eliminate cloud shadows and cloud edge effects. The ECOSTRESS Level-2 Cloud product consists of an 8-bit flag of cloud mask and cloud tests where bit 4 contains the land/water mask [31], which was also used in this study to select water pixels.

From August 2018 to August 2020, 252 ECOSTRESS images were found to have valid SST data covering the 24 sites in CB, LO, CRE, and FB, according to the criteria outlined below to find matching pairs between ECOSRESS and in situ data.

#### 2.1.3. MODIS

MODIS instruments on the Terra and Aqua satellites provide near-daily global SST data with a nominal resolution of ~1 km. MODIS SST was derived from calibrated at-sensor radiance using a multi-channel non-linear regression algorithm [32], and data quality was determined through 15 quality tests including sensor zenith angle, the difference between brightness temperatures/sensor-derived SST and expected brightness temperatures/reference SST, and so on. These tests classified the SST quality using a value between 0 and 3. A pixel that passed all quality tests was assigned the best SST quality value of 0 [33]. Pixels with SST quality values greater than or equal to 2 were considered as invalid pixels contaminated by clouds or other artifacts. The RMS uncertainties are typically within 0.5 °C for most open-ocean waters and < 1 °C for the Gulf of Mexico coastal waters [20]. In this study, MODIS SST data were obtained from the NASA Goddard Space Flight Center. From August 2018 to August 2020, 225 MODIS images were found to have valid SST data covering one site (LONF1) in FB, according to the criteria outlined below to select matching pairs between MODIS and in situ data.

### 2.2. Evaluation Method

#### 2.2.1. Site Selection

In order to evaluate ECOSTRESS SST data (i.e., the ECOSTRESS LST product) for estuaries, only ground stations away from land were used to avoid potential contamination on the image pixels close to land. The distance to land was selected to be ~1.5-pixel size so possible mixed pixels were discarded.

#### 2.2.2. Image Pre-Processing

The initial georeferencing of ECOSTRESS 70 m pixels contained large errors due to uncertainties of the ISS positioning system. In the ECOSTRESS Level-2 data, a geolocation correction using the Landsat base map was applied. However, when cloud cover prevails in a significant portion of the orbit, it is difficult to match ECOSTRESS image with the Landsat base map automatically, resulting in large uncertainties in ECOSTRESS georeferencing. Therefore, a manual geometric correction of the ECOSTRESS products, based on known features on land (including shorelines) was performed before evaluation of the SST data accuracy.

To select ECOSTRESS data with the highest quality, only pixels with bits 1&0 = 00 in the QC bit flags were used. Even though, residual errors due to scattered clouds and cloud edge effects still remained in such quality-controlled images. These residual errors were corrected using computer codes developed with the Photutils of Python package. Specifically, a median filter with adjustable window size (determined from trial and error) was first used to estimate the two-dimensional background and background RMS noise in each location, resulting in a background image and a corresponding RMS noise image. Then, if a pixel’s value exceeds the background ± 2RMS, the pixel is discarded. The procedure was found to be efficient in removing residual errors due to scattered clouds and cloud edge effects.

#### 2.2.3. Matchup Rules for Remotely Sensed SST and In Situ SST

After determining the ground station sites and pre-processing of the ECOSTRESS images using the steps outlined above, the quality-controlled ECOSTRESS SST at each site was calculated as the average of 3 × 3 pixels centered on the site, with the time difference limited to ±1 h from the in situ SST measurement. A spatial homogeneity test was further used to discard pixels with large spatial variability in the matchup selections. Specifically, if the coefficient of variation (CV = standard deviation/mean) of the 3 × 3 pixels was >0.15, the pixels were not selected to compare with in situ SST data. This is because while in situ SST was from the “point” measurement, ECOSTRESS SST was from an area of 70 × 70 m^2^. Such a homogeneity test was necessary to assure that a point measurement could be used to represent a large area. In this step, the use of CV instead of standard deviation in the homogeneity test was to follow the convention of the ocean color research community [34], and also to account for the fact that SST noise may increase with SST following the square root rule if the noise is “white.” Here the term “noise” refers to SST precision as opposed to accuracy, and in practice it is calculated as the standard deviation of 3 × 3 pixels for each center pixel.

For MODIS SST evaluations using in situ SST data, the same criteria were used to find the matching pairs.

#### 2.2.4. Matchup Rules for Concurrent ECOSTRESS and MODIS

For cross-validation of SST derived from two satellite instruments, images of ECOSTRESS and MODIS collected within 3 h were used. Furthermore, ECOSTRESS pixels were aggregated to MODIS resolution in order to have a direct comparison with MODIS pixels. Specifically, from August 2018 to August 2020, 33 ECOSTRESS and MODIS image pairs were found to cover the FB, which were used for the cross-sensor evaluations. Then, MODIS SST images were used to evaluate ECOSTRESS SST using the method outlined below. The use of MODIS data to evaluate ECOSTRESS data was to complement the ECOSTRESS evaluations based on in situ data. Therefore, in this study, only MODIS data over the FB were used for this purpose.

### 2.3. Statistical Measures

The quality of ECOSTRESS SST was assessed using several statistical measures. Among them, linear regression and coefficient of determination (R^2^) were used to assess the correlation between two datasets, and the absolute deviation (bias) and root mean square difference (RMSD) were used to assess the mean difference and spread of the two datasets. Here, linear regression was not used for cross-sensor comparison because regression statistics can be misleading for datasets with small dynamic ranges [35].

## 3. Results

### 3.1. Comparison between ECOSTRESS SST and In Situ SST

As shown in Table 2, for the daytime and nighttime data combined, high correlations (mean R^2^ = 0.92) are found between ECOSRESS SST and in situ SST for the three estuaries and LO. Using in situ SST as the ground truth, mean bias in the ECOSTRESS SST is −0.88 °C, with mean RMSD of 1.53 °C. The negative bias and <1.0 slope values indicate that ECOSTRESS tends to underestimate SST, which may be related to atmospheric water vapor corrections [36] or thin cloud contamination. Overall, these statistics indicate uncertainties of ECOSTRESS SST are higher than those of MODIS SST [20].

Similar observations can be made from the same datasets after they are partitioned into two groups for daytime (8:00–20:00 local time) and nighttime (20:00–8:00 local time), respectively (Figure 2 and Table 2). Determination coefficients averaged over all regions are 0.91 and 0.93 for daytime and nighttime, respectively. Mean bias and RMSD during daytime for all regions are −0.51 and 1.39 °C, respectively, while mean bias and RMSD during nighttime for all regions are −1.20 and 1.62 °C, respectively.

From these observations, ECOSTRESS SST is negatively biased, and the bias is higher at night than during the day. This may be caused by the fact that while ECOSTRESS senses the surface skin temperature (top millimeters), the in situ SST was measured as a bulk temperature of the top water layer (up to 2 m deep). During the day, solar insolation may result in a higher skin temperature than bulk temperature, while during the night the surface cooling may have an opposite effect, thus causing more negative biases at night than during the day. Such effects have been demonstrated in detail in several previous studies [11,37,38,39,40,41].

The negative bias of the ECOSTRESS SST is also revealed in the monthly mean data (Figure 3). Monthly mean temperature differences (TDs) between ECOSTRESS and in situ SST show negative bias in the former for all regions, and the negative bias appears to be worsened after the transition period when only 3 of the 5 bands were used to estimate SST (Figure 3a–d). When all months are combined, mean TDs before and after the transition period are −0.80 and −1.15 °C, respectively (Figure 3e). This result is consistent with those reported in [31], where RMS error in the 5-band LST was smaller than in the 3-band LST. In contrast, the reduced number of bands after the transition period does not appear to cause more data spread, as mean standard deviations (STDs) before and after the transition period are 1.05 and 0.73 °C, respectively.

The negative bias in ECOSTRESS SST appears to vary seasonally (Figure 4), with autumn and winter showing lower biases (mean TDs = −0.64 °C) than spring and summer (mean TDs = −1.17 °C). The data spread in all seasons is rather similar, with mean STDs = 1.26 °C in spring and summer and mean STDs = 1.01 °C in autumn and winter. This observation is consistent with the statistical measures in Table 2, where a slope of < 1.0 indicates more negative bias for higher SST data.

### 3.2. Comparison between ECOSTRESS SST and MODIS SST

To use MODIS SST to evaluate the quality of ECOSTRESS SST, MODIS SST was first evaluated using in situ SST. In Table 2, the determination coefficient between MODIS and in situ SST for daytime and nighttime data combined is 0.98, with bias and RMSD for MODIS SST being 0.11 and 0.51 °C, respectively. When data are partitioned to daytime and nighttime, bias in MODIS SST appears to be higher during daytime (0.19 °C) than during nighttime (0.05 °C) (Figure 5), possibly due to the effect of diurnal heating of the sea surface, and such an effect was also observed in the different biases in the ECOSTRESS daytime and nighttime SST data as explained above. Nevertheless, an RMS uncertainty of ~0.5 °C and bias < 0.2 °C suggest that the MODIS SST is accurate and therefore can be used to evaluate the ECOSTRESS SST.

From 33 ECOSTRESS and MODIS image pairs between August 2018 and August 2020, mean bias and RMSD were calculated for ECOSTRESS SST. For daytime and nighttime data combined, mean bias and RMSD in ECOSTRESS SST are −0.92 and 1.24 °C, respectively. For daytime data, they are −0.76 and 1.24 °C, respectively. For nighttime data, they are −1.05 and 1.24 °C, respectively. ECOSTRESS SST is lower than MODIS SST for both daytime and nighttime, with a smaller difference in their corresponding negative biases than the difference between ECOSTRESS daytime and nighttime biases when in situ SST data were used as the reference. This is because both ECOSTRESS and MODIS essentially detect the same water “skin”, thus would respond similarly to diurnal warming. However, the negative bias in ECOSTRESS SST appears to be larger in spring and summer (mean bias = −1.33 °C) than in autumn and winter (mean bias = −0.49 °C) (Figure 6).

Comparisons between ECOSTRESS and MODIS SST from the 33 image pairs are presented in Figure 7a for daytime SST and Figure 7b for nighttime SST. ECOSTRESS SST covers a range of 18.01–34.72 °C and 17.58–34.47 °C during daytime and nighttime, respectively, while MODIS SST covers a range of 19.93–33.36 °C and 19.13–32.25 °C. From these density plots, it is clear that ECOSTRESS SST is systematically underestimated for all SST ranges, especially when SST exceeds ~25 °C during daytime. On the other hand, the negative bias is mostly small and can be corrected using linear regressions.

## 4. Discussion

Given that MODIS can provide ~1600 images (both daytime and nighttime) per year over the South Florida estuaries while ECOSTRESS can provide only ~160 images per year, the question is what value ECOSTRESS may bring to the table when assessing the thermal environments of these coastal water bodies. The answer is in two aspects. First, for relatively small estuaries (<4 km in either dimension), there are usually no valid MODIS pixels, and ECOSTRESS can fill the data gap. Likewise, in estuaries where MODIS does provide valid SST pixels, these pixels cover waters 1–2 km away from land, and the coarse resolution often smears horizontal SST gradients. In contrast, ECOSTRESS covers waters as close as 100 m from land, and the 70 m resolution pixels reveal more detailed SST changes than enabled by MODIS. These effects are demonstrated using the FB examples below in Figure 8 and Figure 9, as MODIS shows no valid SST pixels over the CRE.

From the 33 concurrent (< 3 h) image pairs, the number of valid SST pixels in each location is shown in Figure 8a for MODIS (1 km) and Figure 8b for ECOSTRESS (70 m), respectively. The frequency of valid SST observations is higher for ECOSTRESS, as its finer resolution enables more observations between small clouds. This effect has been shown in [42], where MODIS 250 m bands were shown to have 10–25% more valid coverage over the Gulf of Mexico than MODIS 1-km bands. Furthermore, in waters close to land, only ECOSTRESS can provide valid SST data (Figure 8c). For the FB region shown in Figure 8c, ECOSTRESS provides 18% more spatial SST coverage than MODIS.

The finer resolution also enabled ECOSTRESS to capture more spatial features than being offered by MODIS, as demonstrated in the image pair of Figure 9a,b. In these images, ECOSTRESS shows more spatially coherent SST patterns, for example, the SST fronts south of the Florida Keys (arrows in Figure 9b). However, the ECOSTRESS SST data in Figure 9b shows distinct striping noise in the along-scan direction. This type of striping noise is typical in either scanning (east–west) or push broom (north–south) satellite sensors, and can be reduced in post-processing [43,44,45]. Some of the noise is due to damaged detectors in TIR bands 1 and 5 and SWIR band during the pre-launch test, resulting in the loss of 8 rows of data per 128 rows in the cross-track direction in these bands [46]. Such striping noise is masked through the use of the QC bit flags (color coded white in the image), and was not used in any analysis above. Other striping noise is caused by the different detector responses [46], which may cause unsmooth pixel-to-pixel SST changes in the along-track direction. Therefore, the image-based anisotropic variograms were derived to show the contrast between the two directions (along-track and along-scan) and the contrast between ECOSTRESS and MODIS images (Figure 9c,d). Here, based on spatial autocorrelation, variograms are used to describe spatial scales of variability (or “patchiness”) as well as the proportion of the variability captured at a given resolution [47,48]. The semivariance is obtained by calculating half of the squared difference of all pixel pairs that are at a distance from each other in any direction in an image. In such derived semivariance parameters, the ratio of minor range (A_min_) to major range (A_max_) between 0 and 1 is used to represent spatial anisotropy, with smaller ratios representing higher spatial anisotropy. A ratio of 1 corresponds to the isotropic case [49]. The ratio is 0.70 for MODIS and 0.39 for ECOSTRESS (Figure 9c,d), indicating higher spatial anisotropy in ECOSTRESS SST, apparently due to the striping noise. The nugget (C_0_) represents the unresolved variability at a given resolution that can be due to noise in SST data, and sill (C + C_0_) is the total variability of a region. From Figure 9c,d, C/(C_0_ + C) is 0.89 for MODIS SST and 0.79 for ECOSTRESS SST, suggesting 89% of structural and 11% non-structural variability in the MODIS SST image, and 79% of structural and 21% of non-structural variability in the ECOSTRESS image. The higher non-structural variability in the ECOSTRESS image is apparently due to the striping noise. On the other hand, ECOSTRESS appears to have higher semivariance at a large separation distance (e.g., >10 km), which is speculated to be caused by the periodic noise in ECOSTRESS. However, the above observations are based on one image pair only, and future studies may use more image pairs to make more statistically meaningful observations.

With a finer spatial resolution, compared to MODIS, ECOSTRESS SST appears to be able to resolve more details in small SST features. When evaluated using statistics of 3 × 3 pixels from homogenous scenes over clear waters, average noise in ECOSTRESS is only 0.13–0.17 °C from 3 images, as compared with MODIS SST noise of 0.10–0.19 °C. This suggests that although ECOSTRESS SST uncertainties (relative to ground “truth”) are higher than MODIS SST, their precisions are comparable regardless of the resolutions.

Finally, even though ECOSTRESS SST can fill data gaps in small estuaries and reveal more spatial features in other estuaries, the uncertainties are higher than MODIS. While the RMS uncertainties may be reduced after pixel averaging, the small negative bias may be corrected through empirical regression. Even without correction, as long as the bias is systematic rather than random, it will not impact the assessment of thermal anomalies. Furthermore, although the assessment is focused on South Florida waters, results from CB (Table 2 and Figure 2) suggest that the observations are similar for temperature ranging from ~6 to ~32 °C, therefore making ECOSTRESS SST data possibly applicable to most estuaries in North America with a similar temperature range.

## 5. Conclusions

While the ECOSTRESS sensor was designed for terrestrial use (e.g., vegetation water stress, agricultural vulnerability, etc.), the evaluation of the ECOSTRESS LST data product (used as a surrogate for SST) over several estuaries and Lake Okeechobee suggests that it is also applicable for these relatively small water bodies for a large temperature range. Such a finding is based on the evaluations of 1076, 287, 331, and 245 ECOSTRESS SST images covering the Chesapeake Bay, Lake Okeechobee, Caloosahatchee River Estuary, and Florida Bay, respectively, from August 2018 to August 2020. The residual errors in georeferencing and negative bias, as revealed by in situ and MODIS SST, can all be corrected through post-processing. Therefore, ECOSTRESS SST may serve as a valuable data source to evaluate the thermal environments of coastal and inland waters.

## Figures and Tables

**Figure 1 sensors-21-04341-f001:**
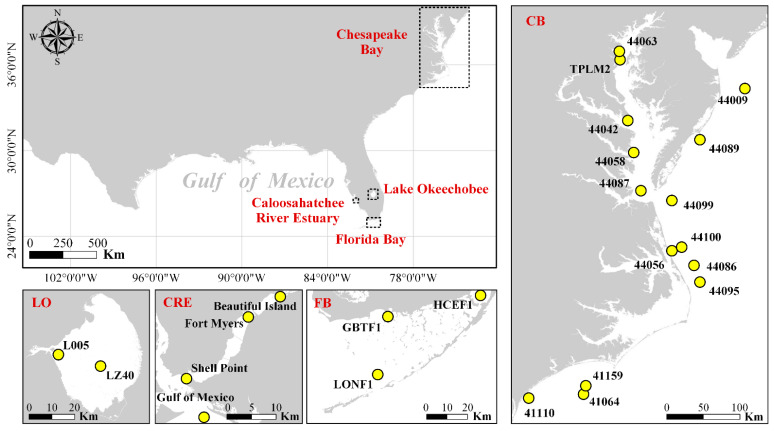
Geographic distribution of the in situ SST measurement sites in three major estuaries and Lake Okeechobee. Detailed information on each site is listed in Table 1.

**Figure 2 sensors-21-04341-f002:**
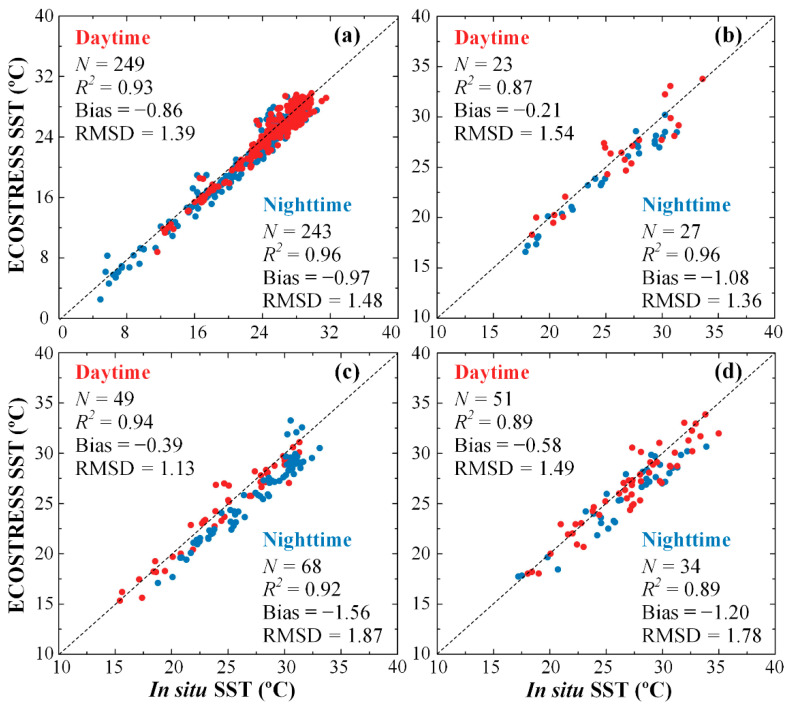
Comparison between in situ SST and ECOSTRESS SST in (**a**) CB, (**b**) LO, (**c**) CRE, and (**d**) FB from the matching pairs found between August 2018 and August 2020. The dotted lines are the 1:1 lines. Daytime and nighttime data are color-coded in red and blue, respectively.

**Figure 3 sensors-21-04341-f003:**
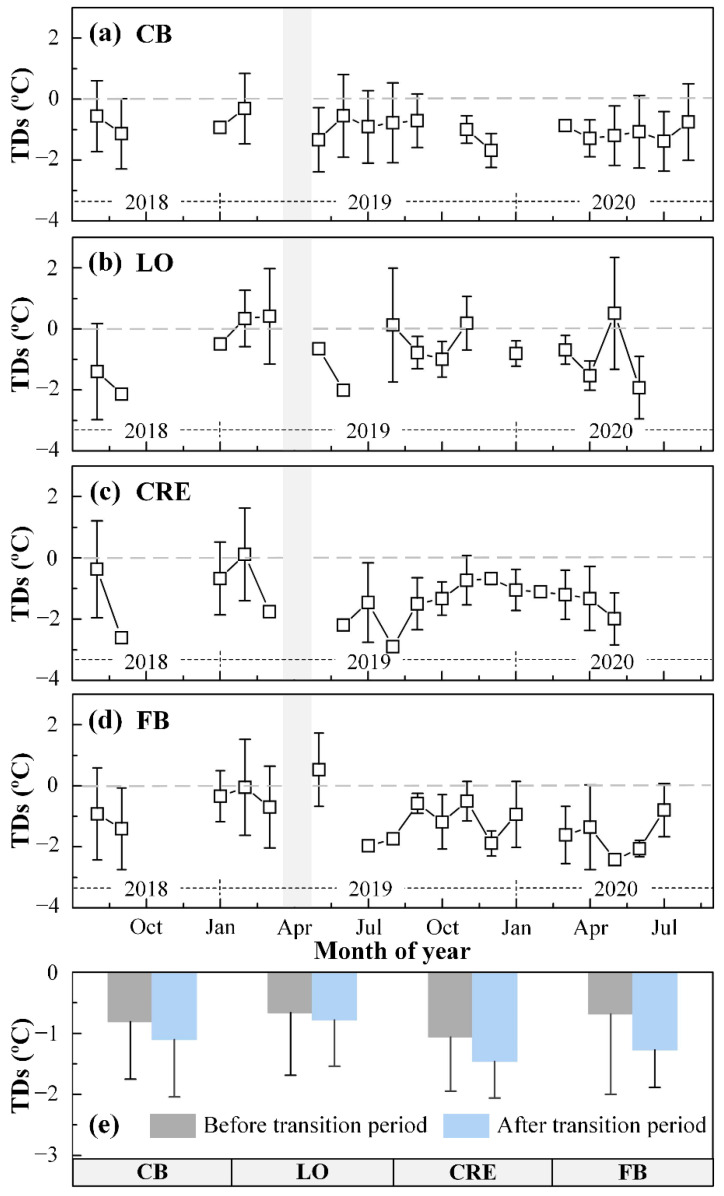
Mean monthly temperature differences (TDs) and standard deviations (STDs) between ECOSTRESS SST and in situ SST from the matching pairs found between August 2018 and August 2020 for (**a**) CB, (**b**) LO, (**c**) CRE, and (**d**) FB. The gray bars indicate the transition period when ECOSTRESS changed its data collection method. Data gaps are due to lack of matchup data pairs; The summary of TDs and STDs for each estuary or lake is shown in (**e**), with data separated by the transition period.

**Figure 4 sensors-21-04341-f004:**
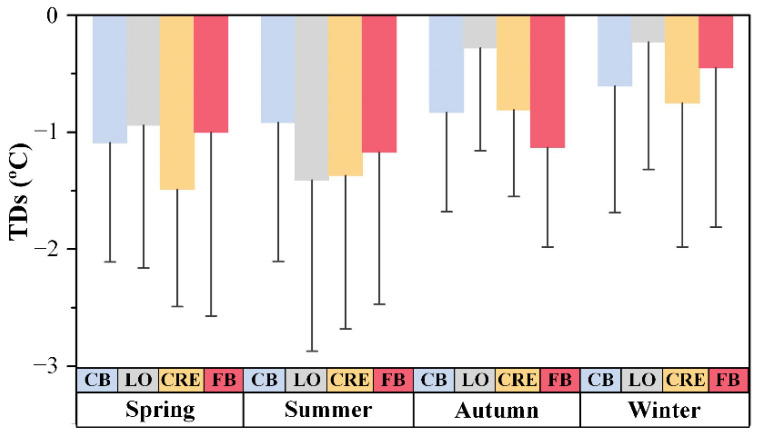
Seasonally mean temperature differences (TDs) and standard deviations (STDs) between ECOSTRESS SST and in situ SST from August 2018 to August 2020. The blue, gray, yellow, and red bars represent CB, LO, CRE, and FB, respectively.

**Figure 5 sensors-21-04341-f005:**
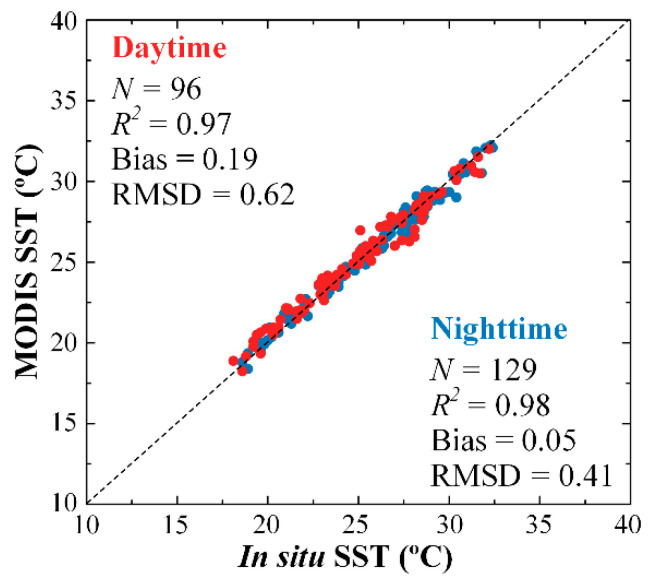
Comparison between in situ SST and MODIS SST in FB from August 2018 to August 2020. The dotted line is the 1:1 line. Data are color-coded for daytime (red) and nighttime (blue).

**Figure 6 sensors-21-04341-f006:**
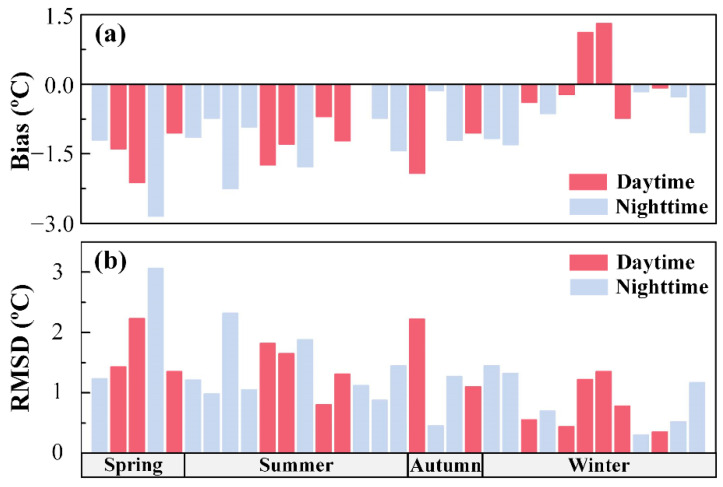
(**a**) Bias and (**b**) RMSD of ECOSTRESS SST against MODIS SST in FB between August 2018 and August 2020 for each season. Data are color-coded in red for daytime and blue for nighttime.

**Figure 7 sensors-21-04341-f007:**
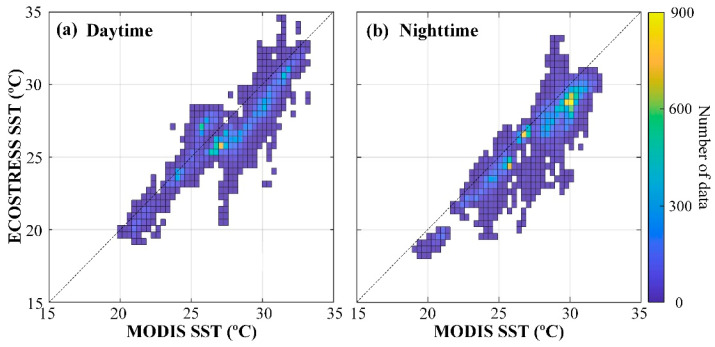
Comparisons between gridded ECOSTRESS SST (to match MODIS pixels) and MODIS SST from 33 image pairs over the FB between August 2018 and August 2020 for daytime (**a**) and nighttime (**b**) data. Color represents the number of pixels from both ECOSTRESS and MODIS.

**Figure 8 sensors-21-04341-f008:**
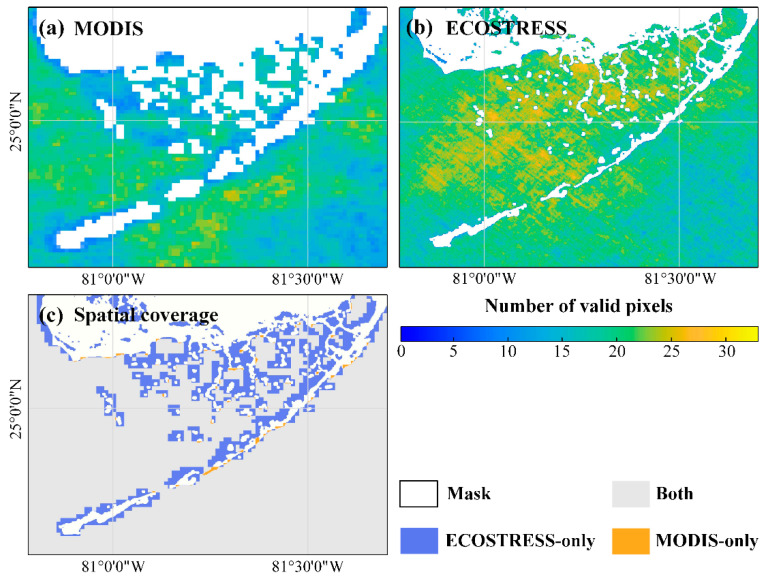
Number of valid pixels from 33 concurrent image pairs of (**a**) MODIS and (**b**) ECOSTRESS from August 2018 to August 2020. Note the resolution difference between MODIS (1 km) and ECOSTRESS (70 m); (**c**) Spatial coverage of the 33 concurrent MODIS and ECOSTRESS images between August 2018 and August 2020. White color represents land and masked pixels, and gray color indicates valid pixels from both ECOSTRESS and MODIS.

**Figure 9 sensors-21-04341-f009:**
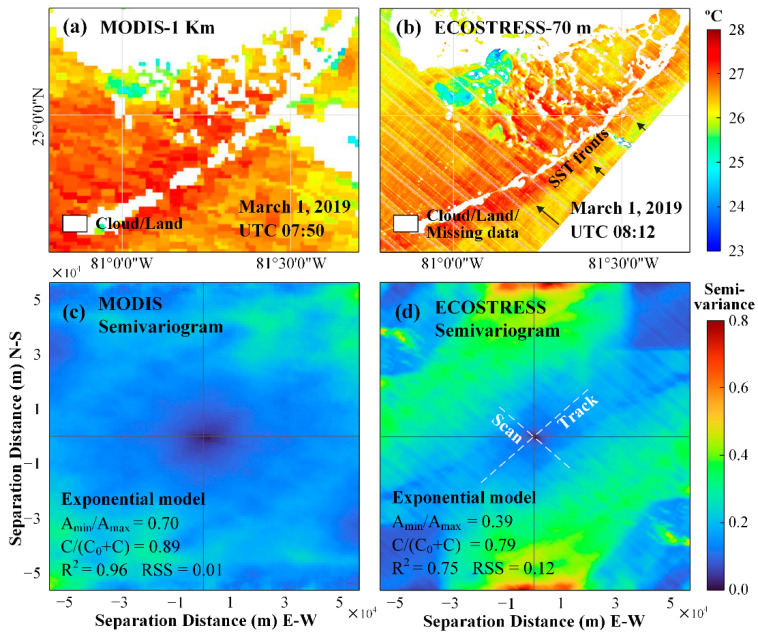
(**a**) MODIS 1-km resolution SST image at 7:50 UTC on 1 March 2019; (**b**) ECOSTRESS 70 m resolution SST image at 8:12 UTC on 1 March 2019. White color represents land, cloud, and masked pixels due to missing data. SST fronts are annotated with arrows in (**b**); (**c**) Two-dimensional semivariogram of MODIS SST in (**a**); (**d**) Two-dimensional semivariogram of ECOSTRESS SST in (**b**). Note that the images are symmetric between negative and positive axis labels. The gray solid lines indicate the north–south and east–west directions, and the white dashed lines represent the along-track and along-scan directions, respectively. SST semivariance is color coded to the right. Among the estimated parameters from the anisotropic exponential semivariogram model, A_min_ and A_max_ represent minor range and major range, and C_0_ and C_0_ + C represent nugget and sill, respectively.

**Table 1 sensors-21-04341-t001:** Information of sites used in this paper, where the bottom depth was derived from an NOAA bathymetry database (100 m resolution). The exact depths of the thermistors have not been reported but are between 0.5 and 2 m below the water surface.

Site Name	Area	Location	Type	Time Range	Time Interval (h)	Bottom Depth (m)	Sources
41064	CB	34.21° N, 76.95° W	Moored buoy	2015.06-	1.0	30.3	NDBC
41159	CB	34.21° N, 76.95° W	Waverider buoy	2015.08-	0.5	30.3	NDBC
41110	CB	34.14° N, 77.72° W	Waverider buoy	2008.05-	0.5	17.6	NDBC
44095	CB	35.75° N, 75.33° W	Waverider buoy	2012.04-	0.5	19.3	NDBC
44086	CB	36.00° N, 75.42° W	Waverider buoy	2018.08-	0.5	21.5	NDBC
44100	CB	36.26° N, 75.59° W	Waverider buoy	2008.05-	0.5	25.8	NDBC
44056	CB	36.20° N, 75.72° W	Waverider buoy	2007.12-	0.5	16.8	NDBC
44099	CB	36.91° N, 75.72° W	Waverider buoy	2008.07-	0.5	21.0	NDBC
44087	CB	37.03° N, 76.15° W	Waverider buoy	2018.08-	0.5	8.8	NDBC
44058	CB	37.57° N, 76.26° W	Moored buoy	2008.11-	0.2	7.6	NDBC
44089	CB	37.75° N, 75.33° W	Waverider buoy	2016.06-	0.5	17.7	NDBC
TPLM2	CB	38.90° N, 76.44° W	C-MAN station	1985.10-	1.0	4.0	NDBC
44063	CB	38.96° N, 76.45° W	Moored buoy	2010.05–2020.07	0.2	6.8	NDBC
44009	CB	38.46° N, 74.70° W	3 m discus buoy	1984.01-	1.0	27.0	NDBC
44042	CB	38.03° N, 76.34° W	Moored buoy	2007.09–2020.06	1.0	13.4	NDBC
L005	LO	26.96° N, 80.97° W	Platform-based station	2020.05-	0.3	2.7	SFWMD
LZ40	LO	26.90° N, 80.79° W	Platform-based station	1990.04–2020.08	0.5	4.3	SFWMD
Gulf of Mexico	CRE	26.44° N, 81.97° W	Moored buoy	2007.11–2020.02	1.0	4.9	SCCF RECON
Shell Point	CRE	26.52° N, 82.01° W	Moored buoy	2008.01-	0.2	0.7	SCCF RECON
Fort Myers	CRE	26.65° N, 81.88° W	Moored buoy	2007.12-	1.0	2.4	SCCF RECON
Beautiful Island	CRE	26.70° N, 81.81° W	Moored buoy	2012.11-	1.0	1.1	SCCF RECON
LONF1	FB	24.84° N, 80.86° W	C-MAN station	1992.12–2020.08	1.0	2.7	NDBC
GBTF1	FB	25.17° N, 80.80° W	Water quality station	2015.05-	1.0	0.7	NDBC
HCEF1	FB	25.25° N, 80.44° W	Water quality station	2015.05-	1.0	0.3	NDBC

**Table 2 sensors-21-04341-t002:** Statistics of the matchup pairs between in situ SST and ECOSTRESS/MODIS SST from August 2018 to August 2020. Here, “Area” represents estuary or lake name. The time column indicates whether the data is collected in the daytime, or nighttime, or a combination of both (local time). *N* is the number of matching pairs, R^2^ is the coefficient of determination, slope and Intercept are the linear regression coefficients between the two data sets, bias represents the mean absolute deviation in the ECOSRESS/MODIS data, and RMSD is the root mean square difference.

Satellite	Area	Time	*N*	R^2^	Linear Regression		Bias	RMSD
					Slope	Intercept		
ECOSTRESS	CB	All	492	0.96	0.99	−0.75	−0.92	1.43
		Day	249	0.93	0.99	−0.69	−0.86	1.39
		Night	243	0.96	0.99	−0.75	−0.97	1.48
	LO	All	50	0.91	0.95	0.72	−0.68	1.45
		Day	23	0.87	0.93	1.61	−0.21	1.54
		Night	27	0.96	0.93	0.58	−1.08	1.36
	CRE	All	117	0.92	0.93	0.77	−1.07	1.61
		Day	49	0.94	0.94	1.06	−0.39	1.13
		Night	68	0.92	0.99	−1.41	−1.56	1.87
	FB	All	85	0.88	0.88	2.33	−0.83	1.61
		Day	51	0.89	0.90	2.11	−0.58	1.49
		Night	34	0.89	0.85	2.87	−1.20	1.78
MODIS	FB	All	225	0.98	0.94	1.60	0.11	0.51
		Day	96	0.97	0.92	2.21	0.19	0.62
		Night	129	0.98	0.96	0.99	0.05	0.41

## Data Availability

ECOSTRESS data and MODIS data used in this research are from the NASA Jet Propulsion Laboratory (https://ecostress.jpl.nasa.gov) and NASA OB.DAAC (https://oceancolor.gsfc.nasa.gov) respectively. In situ SST data used in this research are from NOAA NDBC, SFWMD, and SCCF RECON.

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
