# Peer review of "Evaluation of ECOSTRESS Thermal Data over South Florida Estuaries"

_sensors, 2021, doi:10.3390/s21134341_

Round 1

Reviewer 1 Report

This is a data/SST validation study over estuaries. Nothing technically wrong. Just a few things the authors need to clarify.

(1) I do not understand what the authors means uncertainties in ECOSTRESS data. Based on descriptions in the manuscript, LST/SST data from ECOSTRESS have been evaluated in some forms as provided by retrieval error but I am not sure why such an evaluation is not enough for sea surface/estuaries.

(2) THe ECOSTRESS data seems to have stripping issues (as shown in Fig. 9). Why does the image have the "noise" lines and would these lines affect your evaluation results? What is the semi-variogram for?

(3) Even though MODIS LST data may be good, it does not mean ECOSTRESS and MODIS LSTs vary in the same direction (sign). Thus, you can still see large variations between them. For example, if MODIS is relatively higher than in situ LST while ECOSTRESS SST is smaller than in situ LST, you may see large differences between the two while they both are relatively accurate. Can you identify these cases as well in your valiation?

Reviewer 2 Report

The paper deals with an assessment of ECOSTRESS  SST estimates in estauries insouth Florida. ECOSTRESS was not designed to provide insights of SST over sea, but the results of the Authors demonstrate the feasibility to use ECOSTRESS estimates in very coastal area, and this could be of great benefit to gaining insights into estauries inSouth Florida and possibly in other very coastal areas. The paper is weel written, clear and I enjoiy reading it.

Only very minor comments from me

  • lines 58 and 62: please use Authors's name instead of the number reference ;
  • line 85: use "lesser than" instead of "<"   

is well written, clear and I enjoy reading it.

Reviewer 3 Report

The objective of the work reported in this manuscript is an evaluation of the infrared sensor ECOSTRESS flown on the Space Station. SST retrievals obtained from ECOSTRESS are compared with those from MODIS and with in situ values. As the authors argue, the significantly higher resolution of ECOSTRESS retrievals, compared with those from MODIS, make these data of particular interest for coastal studies, rendering such a study of value. However, I believe that the analysis requires significant work prior to being accepted for publication. There are three general areas, which I believe need to be addressed:

1) Diurnal warming. This is mentioned briefly but my sense is that it may play a significant role in the comparisons. This is an area that has seen a significant amount of research over the past 10+ years, with many papers showing the effect and models designed to compensate for it. Given its importance, especially in coastal waters, more detail is needed with regard to the in situ measurements. Specifically at what depth are the thermistors on the different platforms - diurnal warming results in significant stratification of the upper meter or so of the water column. Sensor depths could be shown in Table 1. The ECOSTRESS/in situ differences might be examined as a function of sensor depth. I would also have examined these differences as a function of water depth. 

Diurnal warming is affected by solar insolation and by surface wind stress. It would be useful to analyze the results based on wind speed. Some of the in situ measurements are made in areas in which one would expect relatively weaker winds, such as in the Caloosahatchee River Estuary and in these regions one might expect more frequent diurnal warming events. The data tend to bare this out with a clear nighttime/daytime difference in bias for the CRE stations. A more thorough analysis of the more open regions might make use of wind speed analyses available from the National Weather Service. By contrast, MODIS and ECOSTRESS are measuring the same parameter, the temperature of the top few micrometers of the water column so these measurement should be more consistent. Furthermore, as the authors have done, the ECOSTRESS retrievals can be averaged to the MODIS footprint to eliminate issues related so the different areas of water from which the measurements come - a significant issue when comparing in situ to ECOSTRESS retrievals. 

The reason that I emphasize the importance of better characterizing the contribution of diurnal warming to the statistical comparisons is that, for the regions being analyzed, diurnal warming may play a significant role in both the biases and standard deviations of the comparisons. The authors have the data to quantify the effect of diurnal warming on the retrieved uncertainties.

2) Striping.  Apparent in the SST fields shown in both Figs. 8 and 9 is significan striping. I’m guessing that this is a scan-line calibration issue. Given how regular it appears to be - entire scan lines being warmer/cooler than neighboring lines - I’m a little surprised that some effort was not made to remove it. Striping is also a problem for MODIS, at least it is for the analyses I do, but correcting it is very hard because the striping does not occur uniformly on the entire scan but rather on relatively short segments along a given scan line. But I have no experience with ECOSTRESS so correcting it may be a problem here as well. The bottom line, though, is that striping is clearly contributing to significant pixel-to-pixel noise in these data, which will, of course, impact the statistical comparisons with in situ values. The authors make an attempt to estimate the instrument noise using the variogram. However, they determine the variogram by averaging over all directions. I would suggest performing this analysis separately in the along-track and along-scan directions in an effort to quantify the impact of variability in line-to-line calibration. 

In addition to the two issues raised above I have made a number of editorial as well as slightly more significant comments in the attached copy of the manuscript. 

I was also surprised about the sparsity of the references. As noted above, there has been a lot of work performed related to both diurnal warming and striping, issues important in a statistical comparison of datasets of this type. There have also be a number of analyses of satellite SST datasets, comparing one retrieval versus another and/or retrievals versus in situ observations. I would have expected to see reference to one or more of these both as examples of approaches previously used in the SST community to evaluate the quality of the SST fields as well as more direct comparisons of the biases and standard deviations obtained here for MODIS relative to in situ and those obtained by others. (I hope that I didn’t miss such a comparison in the manuscript.)

Round 2

Reviewer 3 Report

The authors have addressed most of the issues I raised in my original review. In doing so they have introduced a few new items, which I think still need to be addressed but they are very close. Also, I have to apologize to them, I have raised a concern about what I feel is extraneous material related to the retrieval of land SST  that I did not mention in the first go-around. I think that it will be very easy to deal with this.

My comments are in the attached manuscript and summarized below:

Notes in ‘sensors-1227293-peer-review-v2’

Notes in Document 

'sensors-1227293-peer-review-v2':

Comment: Is this relevant to the work being described in the manuscript? If not, I would get rid of it in that it adds confusion to the presentation. In fact much of the discussion in this and the next paragraph appear to be related to land retrievals, which I found to be confusing. If they are important, you need to say why. For example, is the cloud shadow information mentioned in the next paragraph used?
(sensors-1227293-peer-review-v2, p.3)

Comment: Not sure what you mean by ‘fill holes’ here. Is this part of the cloud test or were single or a small number of pixels flagged as cloudy replaced somehow with SST values?
(sensors-1227293-peer-review-v2, p.3)

Comment: I wonder if ‘scattered’ might be better than ‘broken’?
(sensors-1227293-peer-review-v2, p.5)

Comment: Same comment ‘broken’ —> ‘scattered’
(sensors-1227293-peer-review-v2, p.5)

Comment: I don’t understand this. Are you referring to the noise introduced by the non-linear term? By noise here are you referring to the accuracy or the precision?
(sensors-1227293-peer-review-v2, p.6)

Comment: I’m confused here. Both  ECOSTRESS and MODIS SSTs are derived from infrared observations both of this are just seeing the surface so, unless a specific correction was made to the data of one to match it to a bulk SST there should be no skin temperature bias of one instrument relative to the other. This is even true if the algorithm for one is tuned to a skin temperature and the other to a bulk temperature. Tuning in this regard would in an overall bias of the signal but not a bias at day compared to night. I’m sorry if I have missed something here.
(sensors-1227293-peer-review-v2, p.10)

Comment: I’m not sure how the coefficients for the  ECOSTRESS algorithm are determined but I believe that those for MODIS are determined from matchups (globally) for a period of time, such as 5 months. If the ECOSTRESS coefficients are determined for the length of the mission, then one might see seasonal biases when comparing the two.
(sensors-1227293-peer-review-v2, p.10)

Comment: Are you suggesting that ECOSTRESS captures more geophysical variability for separations larger than, say, 10 km than MODIS? This comes as a surprise since the difference in the spatial resolution of the sensors is unlikely to play a role at these scales. I think that I’m missing something here.
(sensors-1227293-peer-review-v2, p.13)

Comment: Estuaries in the winter especially in Canada will have temperatures down to 0C. This statement seems to a bit of an oversell.
(sensors-1227293-peer-review-v2, p.14)

Comment: I’d get rid of ‘slight’. It is a subjective term and, compared to what I have seen for MODIS, the biases don’t seem slight to me.
(sensors-1227293-peer-review-v2, p.15)
